# The Influence of Blood Parameters on the Adhesion of an Epidermal Substitute in the Treatment of Burn Wounds in Children

**DOI:** 10.3390/jcm14134614

**Published:** 2025-06-29

**Authors:** Aleksandra Barbachowska, Piotr Tomaka, Agnieszka Surowiecka, Maciej Łączyk, Zofia Górecka, Adam Stepniewski, Anna Chrapusta, Rafał Sadowy, Jerzy Strużyna, Tomasz Korzeniowski

**Affiliations:** 1East Center of Burns Treatment and Reconstructive Surgery, 21-010 Lęczna, Poland; dr.surowiecka@gmail.com (A.S.); laczyk.maciej@gmail.com (M.Ł.); zfgrecka@gmail.com (Z.G.); adam.stepniewski@med.uni-goettingen.de (A.S.); jerzy.struzyna@gmail.com (J.S.); 2Department of Plastic, Reconstructive Surgery and Burn Treatment, Medical University of Lublin, 20-093 Lublin, Poland; 3Department of Anesthesiology and Intensive Care, District Hospital, 21-010 Leczna, Poland; p.tomaka@szpital.leczna.pl; 4Department of Plastic and Reconstructive Surgery and Microsurgery, Medical University of Lublin, 20-093 Lublin, Poland; 5Division of Plastic Surgery, Department of Trauma Surgery, Orthopedics and Plastic Surgery, University Medical Center Goettingen, 37075 Goettingen, Germany; 6Malopolska Burn and Plastic Surgery Center, Ludwik Rydygier Memorial Hospital in Krakow, 31-820 Krakow, Poland; anna.chrapusta@gmail.com; 7Faculty of Medicine, Medical University of Lublin, 20-093 Lublin, Poland; rafal.sadowy@gmail.com

**Keywords:** children burn, skin substitute, wound healing

## Abstract

**Background:** Burns in children represent a significant public health issue, as there is no single targeted dressing for the treatment of burn wounds in children. The alloplastic epidermal skin substitute is the dressing of choice for treating burns in children in our burn center. However, it sometimes occurs that the dressing separates from the wound too early, before the process of full re-epithelialization. The inflammatory phase of wound healing seems to be crucial for maintaining the adhesion of the dressing, and thus, changes in parameters such as leukocyte levels and protein changes are of clinical significance. The aim of our study is to find laboratory factors that could contribute to premature dressing separation. **Methods:** The documentation of 182 children treated for acute burns at a major Polish burn center in the years 2009–2023 was analyzed. A demographic analysis was performed to collect information. The group was split into the following two categories based on the condition of the dressing: “attached to the wound” and “detached from the wound”. Laboratory tests were collected on admission and with control tests 3–5 days after injury. **Results:** The results indicate that only a few of the parameters studied showed a statistically significant difference between the groups of patients in whom the dressing did or did not attach. The most pronounced relationship was found for the pre-treatment leukocyte level (leuk1). Statistical significance was also demonstrated for hemoglobin levels and changes in protein (protein_diff) and also glucose levels (glucose_diff). **Conclusions:** Our study shows that there are blood parameters (leukocyte, protein, and glucose levels) that influence the adhesion of the dressing. Unfortunately, there are no other studies on this topic in the literature, so it seems very important to expand research in this direction.

## 1. Introduction

Burns in children represent a significant public health issue, ranking fifth among the most common non-fatal injuries worldwide. Their high incidence highlights the need for effective prevention and treatment strategies [1]. In low- and middle-income countries, burns are one of the leading causes of lost disability-adjusted life years (DALYs) [1]. The specific characteristics of this age group are influenced by their physiology and anatomy. This is largely due to their immune system, which is not yet fully developed, resulting in a higher rate of complications, such as infections and sepsis. Additionally, children have an unfavorable body surface area-to-mass ratio, making them more prone to hypothermia. Their skin is thinner than that of adults, increasing the likelihood of third-degree burns [2]. Children are also particularly vulnerable to burns due to their impulsiveness, lack of awareness of dangers, high activity levels, curiosity, and dependence on caregivers. As a result, they may experience both acute, life-threatening complications and long-term consequences, such as disfigurement, disability, and psychological trauma [1,3]. The most common type of burns in children is partial-thickness burns. These incidents most frequently occur at home and are caused by hot liquid spills. Significant advancements in pediatric burn treatment over the past decades—including early fluid resuscitation, effective infection control, and precise wound management—have contributed to a reduction in mortality [4,5]. Careful wound care, involving cleaning, infection prevention, and maintaining a moist environment that promotes faster healing, also plays a crucial role in treatment success. Burns often lead to severe complications, such as scarring, hypo- and hyperpigmentation, and burn contractures. As a result, patients often require reconstructive procedures to restore bodily function [3]. Early referral to specialized burn treatment centers is essential for achieving optimal treatment outcomes, as delayed admission is associated with longer hospital stays and increased complications [6]. Another very important role is played by choosing the right dressing. It is necessary to take into account that children are more sensitive to pain and are less cooperative; therefore, the number of dressing changes and their painlessness play a key role. In optimizing burn treatment, especially in children with a synthetic epidermal substitute, it presents a promising therapeutic option. According to research, its application enhances patient comfort, supports the healing process, shortens hospital stays, and reduces opioid requirements [7,8]. However, no studies indicate whether the primary dressing is left on the wound until it has healed completely. When it comes off before the wound has healed, which may happen, the healing process is affected. Wound healing can be influenced by many factors that disrupt the process, resulting in abnormal or impaired tissue repair. The most obvious is that malnutrition and protein loss can have a huge impact on wound healing following trauma and surgery. Proper glucose levels support the ability to produce collagen, which is crucial. However, slow wound healing may be related to high glucose toxicity in glucose-induced immune suppression mechanisms. Leukocytes actively participate in the inflammatory response and subsequent phases of wound healing. They are essential in fighting infection, removing debris, and promoting tissue repair [9]. The aim of this study was to find laboratory factors that could contribute to premature dressing separation.

## 2. Materials and Methods

### 2.1. Study Design

The documentation of 182 children treated for acute burns at the East Centre of Burns Treatment and Reconstructive Surgery in Leczna in the years 2009–2023 was analyzed. The demographic data, etiology, and area and degree of the burn, as well as the methods and effects of treatment, were analyzed.

### 2.2. Inclusion and Exclusion Criteria

The inclusion criteria were age between 1 month to 18 years and acute burns up to 5 days after the incident. Individuals who did not meet the inclusion requirements, such as age over 18 years, old burns, and children discharged from the hospital at the request of their parents, were excluded from this study.

### 2.3. Division of the Study Group According to the Severity of Burns

The group was split into two categories based on the condition of the dressing: “attached to the wound” and “detached from the wound”. Laboratory tests, including tests collecting hemoglobin, hematocrit, leukocyte and lymphocyte, protein, C-reactive protein, procalcitonin, and glucose levels, were conducted on admission, and control tests were carried out 3–5 days after injury.

### 2.4. Wound Management

Immediately after the injury, the wounds were protected with a hydrogel dressing by the medical rescue team or in the emergency department. Then, the wound was assessed for the depth and extent of the burn. The debridement procedure was performed no later than the first day after injury under opiate or general anesthesia in the operating theater. This involved thoroughly cleansing the wound and the removal of blisters and keratin remnants.

### 2.5. Dressing Selection

The wounds were covered with an epidermal substitute—Suprathel^®^ dressing (PolyMedics Innovations GmbH, Denkendorf, Germany). Paraffin tulle gas (Jelonet™, Smith & Nephew, Watford, UK) and gauze with Prontosan (B. Braun, Melsungen, Germany) were used as a protective top dressing, followed by bandages and an elastic dressing mesh.

### 2.6. Intervention

The wounds were inspected every 2–3 days, and the top layers of the dressing were replaced until epithelialization occurred. When the dressing was separated from the wound earlier, the wound was covered with a fresh paraffin dressing. If spontaneous healing did not progress in 14 days, the wounds were treated with split-thickness skin grafting. The exception was 3rd degree wounds, where it was decided to immediately remove the necrotic tissue and perform skin transplantation.

### 2.7. Dressing Adhesion Assessment

The dressing was assessed as attached or detached based on clinical assessment by an experienced burn surgeon. Each intervention was documented with an appropriate note on adherence in the surgical protocol and wound photographs.

### 2.8. Laboratory Tests

Laboratory tests, including hemoglobin, hematocrit, leukocyte and lymphocyte, protein, C-reactive protein, procalcitonin, and glucose levels, were collected on admission, and control tests were conducted after 3–5 days after injury.

### 2.9. Characteristics of Dressing Used in This Study

The skin substitute selected and used in this study was Suprathel^®^. Suprathel^®^, a biosynthetic skin substitute, is a microporous, hydrolytically absorbable membrane measuring 70–150 μm, consisting mainly of DL-lactic acid, trimethylene carbonate, and caprolactone [10,11]. Suprathel^®^ mimics epithelial characteristics and possesses semi-occlusive properties. However, due to its high-porosity structure, it enables the wound’s moisture to pass through, which prevents the accumulation of wound fluid, promoting wound healing and the regeneration of the epithelial layer. It also provides significant plasticity for immediate adaptation to wound beds at body temperature [12]. When the healing process of the wound is not disturbed, there is no need for daily dressing changes, preventing secondary mechanical damage to keratinocytes migrating across the wound surface, which is typically caused by removing adhered dressings [8,13].

### 2.10. Ethical Statement

The Declaration of Helsinki’s principles were adhered to. This study received approval by the Institutional Ethics Committee of the Independent Public District Hospital in Leczna (ref. number: 02/WCLO/2023): approval date: 20 March 2023.

### 2.11. Statistical Analysis

Statistical analysis was performed using the Welch *t*-test, which is a modification of the classic Student’s *t*-test, and adjusted to the situation of unequal variances in the compared groups and unequal sample sizes. This test was chosen due to its increased robustness to the violations of the assumptions of homogeneity of variance. The results were presented using mean differences between groups, together with their 95% confidence intervals (CI95%), supplementing the analysis with an estimate of the effect size using the Hedge coefficient (ĝ_Hedges). For multiple testing, Holm corrections were applied. Logistic regression was used to verify the influence of parameters on the probability of dressing adhesion.

## 3. Results

The group’s features are displayed in Table 1.

The patients were divided into two groups—in the first group, the first dressing was an alloplastic skin substitute that adhered to the wound, and in the second group, it was partially adhered or completely came off the wound (Figure 1).

Then, specific blood parameters were analyzed depending on the groups. Analysis of variance was performed to assess the differences in the hemoglobin level variable between the three groups, describing the condition of the dressing as follows: “attached to the wound” and “detached from the wound”. The result of the statistical test presented in the graph is the Welch test (t-Welch), with a test statistic value of 2.48, 163.07 degrees of freedom, and a *p*-value of 0.01 (*p*_adj_ < 0.05), which means that the observed difference in mean hemoglobin levels between the groups is statistically significant. The mean hemoglobin value for patients in whom an epidermal skin substitute did not assist healing is 12.85, while for patients in whom the dressing assisted healing, the mean hemoglobin is 12.41 (Figure 2).

The graph (Figure 3) shows a comparison of the variable leukocyte values between the two groups. When analyzing the graph by comparing the leukocyte values in the two groups of patients with dressings “attached to the wound” and “detached from the wound”, a clear difference between the groups can be seen. Patients in whom the dressing did not attach to the wound have an average higher leukocyte level (mean of about 15.31) compared to patients in whom the dressing attached to the wound. This difference is statistically significant, as confirmed by the very low *p*-value (*p* < 0.01, *p*_adj_ < 0.01). The effect size (ĝ_Hedges = 0.48) indicates a moderate effect, which means that this difference may be significant not only statistically but also clinically. The confidence interval for this difference does not include zero, which additionally supports the belief in the significance of the difference.

Statistical analyses of hematocrit, lymphocyte, protein, C-reactive protein, procalcitonin, and glucose levels did not reveal any statistical correlation with dressing adhesion. In the second part of our retrospective study, we examined the effect of blood parameter differences on dressing adhesion. The studies were repeated 3–5 days after the dressing application. In this part of the study, by analyzing the presented graph and comparing changes in protein levels (protein_diff), i.e., the difference between the values obtained before and after treatment, a statistically significant difference is noticed between both groups. Patients in whom the dressing did not attach to the wound experienced a greater average decrease in protein levels (average change of about −0.57), while in patients with a dressing attached to the wound, the decrease was much smaller (average change of about −0.12). The obtained difference is statistically significant (*p* = 0.02, *p*_adj_ < 0.05), and the effect can be considered moderate (ĝ_Hedges = −0.47). Importantly, the confidence interval for the size of the effect does not include zero, which additionally strengthens the significance of the observed effect. The graph shows individual outliers, especially in the group where the dressing did not heal, which may indicate the presence of patients with an exceptionally large decrease in protein levels (Figure 4).

Analyzing another graph of the change in glucose levels (glucose_diff), which shows the difference in values between the measurement before and after treatment, one can see a statistically significant difference between the analyzed groups. In the group of patients without dressing adhesion, the average change in glucose is clearly higher and amounts to about −42.37, while in the group of patients with proper dressing adhesion, the change is much smaller and amounts to about −5.34. This difference is statistically significant (*p* = 0.03, *p*_adj_ < 0.05), and the effect size is large (ĝ_Hedges = −0.76), which suggests that the observed difference may have significant clinical significance. The confidence interval for the effect does not include the zero value, which additionally confirms the reliability of the observed difference. The graph shows a large variability of values in both groups and the presence of individual outliers, indicating significant changes in glucose levels in some patients, especially in the group without dressing adhesion (Figure 5).

Most of the other variables listed earlier did not show significant differences between groups in the Welch test (*p* > 0.05), and the values of ŋ^2^_p_ indicated no or very small effect differences. These results suggest that the dressing status does not have a significant effect on these laboratory parameters, and any differences may be due to chance and the randomness of the data.

The logistic regression model indicates that, among the analyzed variables, only protein_diff has a potentially significant effect on the dependent variable, reaching an odds ratio (OR) value of 4.29, with a 95% confidence interval from 1.20 to 24.9 and a borderline *p* value of 0.055, which may suggest a trend in significance. The glucose_diff variable shows a small effect (OR = 1.02; 95% CI: 1.00–1.06) and is also close to the level of significance (*p* = 0.090).

## 4. Discussion

When treating burns in children, it is important to take into account both the perception and expression of pain, as well as the thinner structure of the skin, which significantly differs from adults. These two features play a huge role in the appropriate approach to treatment. Children’s skin is considered smoother and softer, with a thinner stratum corneum. Water regulation differs, and the production of natural moisturizing factors and lipids is lower compared to adults. These features make it more sensitive to inflammation and irritation [14,15]. Despite this, no ideal dressing dedicated to treating burn wounds in children has been created [16,17]. The selection of such a dressing in treatment is usually based on operator preference and access to dressings in the burn unit. In our department, this dressing is an epidermal substitute.

The use of an epidermal substitute in treating burn wounds has been described in the literature for several years. Based on these reports, its use brings many benefits, mainly including reducing the opioid demand, the need for further skin grafting, and the length of hospital stay in pediatric patients [8,18,19,20]. Moreover, an epidermal skin substitute presents highly absorbable features when it comes to decreasing the wound’s exudate and generates an anti-microbial barrier, accelerating the healing process. In the context of treating burn wounds in children, its greatest advantage seems to be the lack of need to change the dressing daily; in fact, the dressing is left to heal the wound on its own by changing its outer layers [18,20,21,22,23]. What can be confirmed is based on the research conducted by Blome et al., which proved that the use of Suprathel^®^ provides satisfactory outcomes in treating second-degree burns. In 2021, the results of their prospective study were released. In total, 138 pediatric patients were enrolled in their study, with all wounds completely treated with an epidermal skin substitute without skin grafting [20,24]. Also, in 2018, Wasiak et al. evaluated randomized controlled trials and proved that healing outcomes in partial thickness and superficial burns were significantly more effective when an alloplastic epidermal substitute was used compared to the use of standard dressings [25]. Additionally, the use of an epidermal substitute decreases the need for skin grafting in children with mixed superficial and deep second-degree burns and reduces the number of procedures requiring general anesthesia [7,23].

However, its effectiveness can only be confirmed when the dressing adheres to the wound. Unfortunately, there are no other studies evaluating the effect of the dressing on wound healing when, at some point after application, it separates from the wound before the time of full re-epithelialization, which can lead to consequences in the form of prolonged wound healing and skin grafting [23]. Based on our observations, this usually occurs in the case of large amounts of wound exudate, which causes the dressing to detach from the wound surface. It is worth mentioning that every burn generates local and systemic consequences, including general inflammatory responses. Local tissue damage causes wound exudate to become more abundant and generates a systemic response, e.g., an increase in the level of leukocytes [4]. The prediction of dressing adherence and final wound closure depends on the formation of granulation tissue or epithelialization processes, which can only occur when the wound is maintained at an adequate level. Excessive exudate and the consequent separation of the dressing may be due to excessive protease activity, inflammatory cytokines, or microbial factors degrading the wound–dressing interface. The determination of biomarkers as prognostic factors seems to be a useful tool for the monitoring and modulation of optimal wound healing. The whole process of wound healing consists of several stages: inflammation, migration, proliferation, and remodeling. Each stage can be described by several markers, although none of them are significantly specific to the given stages [26,27,28]. The migration and activation of white blood cells during the inflammatory phase, such as monocytes, neutrophils, and macrophages, are responsible for the creation of an early immune response to prevent wound infection [29,30,31]. For example, activated macrophages secrete TNF-α, which plays a role in early immune responses, regulating the functions of endothelial cells and impacting the vascularization and epithelialization of the wound bed [29,32]. The elevation of IL-33 stimulates macrophages to increase the production of ECM components and generates neutrophil proliferation, which decreases S. aureus colonization [28,29]. For this reason, the inflammatory phase can be considered the most important for maintaining the adhesion of the dressing, including an epidermal substitute. We take into account another factor, which is glucose. Its fluctuations after burns are described in two successive stages. The first one is the “ebb” phase, and the second one is the “flow” phase. However, the phase characterized by hyperglycemia predominates, negatively affecting wound healing, as confirmed in our work in which premature dressing was separated [33]. It has long been known that the key to proper wound healing is proper nutrition with the supply of nutrients, mainly proteins [34,35]. Protein loss leads to severe dysregulation, which has been attributed to factors such as increased endothelial permeability, hypermetabolism with protein catabolism, and the direct destruction of local proteins in the heat-exposed region [36]. Additionally, as in our study, reducing protein levels can lead to dressing separation. In addition to protein levels, it would be worthwhile to expand future research to include additional data assessing nutrition.

### Limitations of This Study

In this single-center research, only an alloplastic epidermal substitute was used, which limits the scope of this study. However, in our burn center, it is the dressing of choice due to its advantages in treating burns in children. A comparison of other dressings that are used in exceptional clinical situations could give unreliable results regarding adhesion. Still, it would be valuable to conduct research on other dressing materials and use larger sample sizes. The retrospective design (2009–2023) may introduce significant temporal bias due to its very long period. Notwithstanding, we have not observed an evolution of practices for the use of epidermal substitutes. The technique introduced 14 years ago has remained the same. In the study group, there was only one case of third-degree burns (0.5%), and 34.5% of burns were second-/third-degree burns; therefore, the obtained results mainly concern the assessment of adhesion in partial-thickness burn wounds and mixed wounds caused by hot fluids (86.8%). However, the distribution of the study group reflects the typical epidemiology of burns in children. Since no mechanistic experiments were performed (e.g., in vitro adhesion assays under varying glucose/protein conditions), the relationship between hyperglycemia/protein loss and dressing adhesion should be considered as speculative.

To summarize the overall analyses presented, the results indicate that only a few of the parameters studied showed a statistically significant difference between the groups of patients in whom the dressing was attached and those in whom it was not. The estimation of white blood cell elevation as a clinical measurement to predict burn wound healing and dressing adhesion outcomes requires further research. Reducing the inflammatory response at an early stage, for example, by using anti-inflammatory drugs, could probably contribute to improving the quality of dressing adhesion and wound healing. On the other hand, those actions may limit the healing process by reducing the concentration of individual cytokines. Anti-inflammatories such as corticosteroids and NSAIDs have complex effects and may have a paradoxical effect on wound healing. While reducing inflammation, they can also interfere with the body’s natural healing processes, potentially delaying or impairing tissue repair. There is a need for a broader definition of burn patients, in which biomarkers such as cytokines (e.g., IL-6, TNF-α, VEGF) or even the elevation of blood cells could be used with higher specificity.

## 5. Conclusions

Since a burn wound can leave scars and contractures that impair the development of children, epidermal and dermal substitutes may indicate a new path for the development of targeted dressing for the treatment of burn wounds in children. However, their effectiveness depends on proper adhesion to the wound surface. Our study shows that there are parameters (leukocyte, protein, and glucose levels) that may influence the adhesion of the dressing. Unfortunately, there are no other studies on this topic in the literature, so it seems very important to expand research in this direction. Given the exploratory nature of our analysis, prospective studies with more specific cohort validation are indicated. The validation and identification of those markers would create a clinical model of treatment to monitor the efficacy of therapy, including the improvement of dressing change ratios and overall assessments.

## Figures and Tables

**Figure 1 jcm-14-04614-f001:**
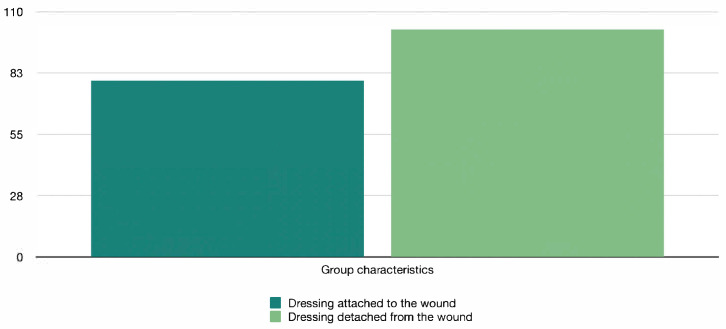
Group characteristics.

**Figure 2 jcm-14-04614-f002:**
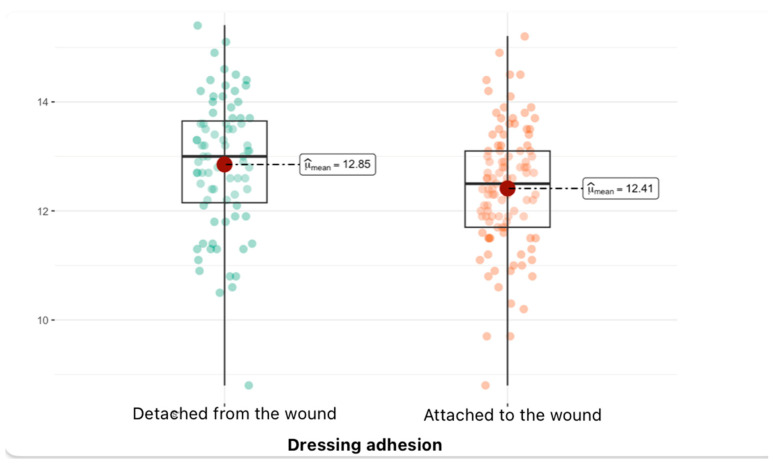
Dependence between hemoglobin and dressing adhesion (median and density distribution).

**Figure 3 jcm-14-04614-f003:**
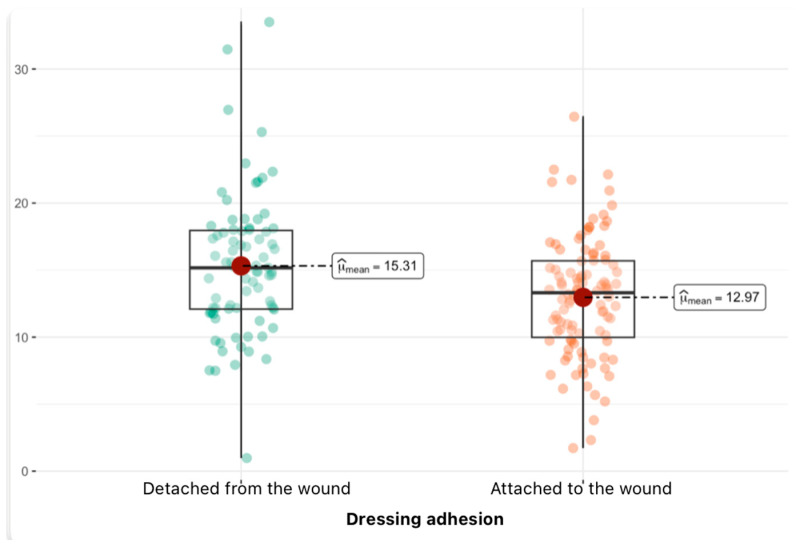
Dependence between leukocytes and dressing adhesion (median and density distribution).

**Figure 4 jcm-14-04614-f004:**
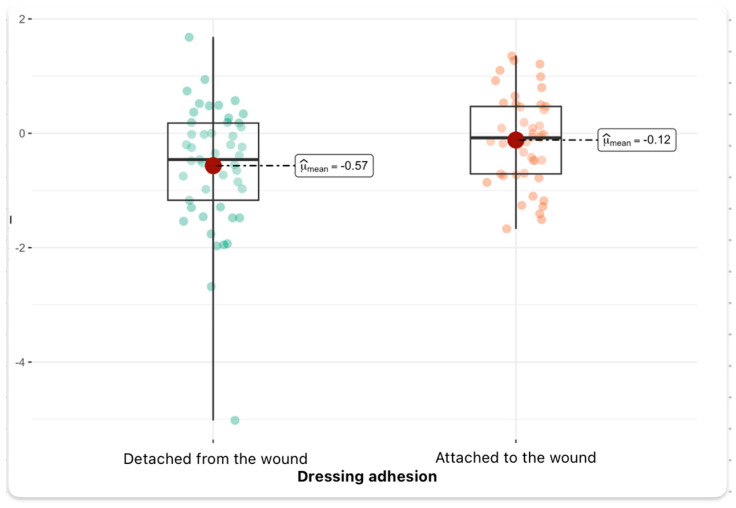
Protein level change depending on dressing condition (median and density distribution).

**Figure 5 jcm-14-04614-f005:**
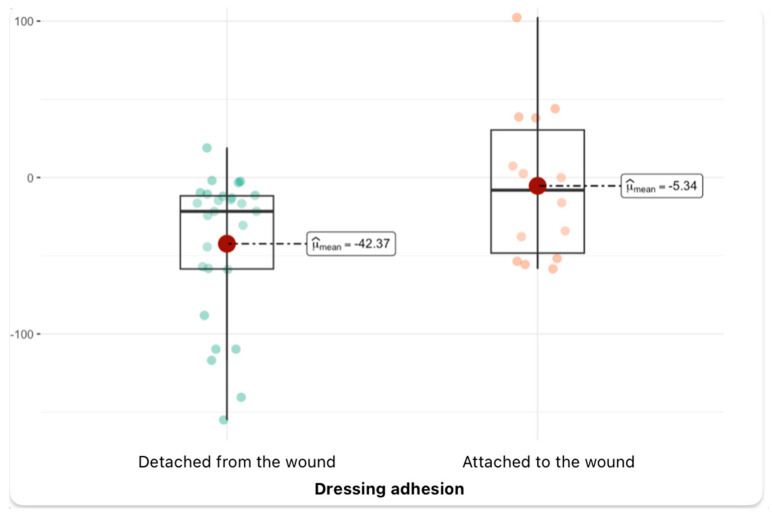
Glucose level change depending on dressing condition (median and density distribution).

**Table 1 jcm-14-04614-t001:** Characteristics of the study group.

	Demographic Characteristics	
Study group	Number of patients	182
	Age (months, mean ± SD/range)	37.143 ± 40.658/166
	Sex (%)	M 62.1%/F 37.9%
Depth of Burn	TBSA Burn % (mean ± SD/range)	10.8 ± 8.73/64.5
	I/II (N/%)	32/17.582%
	II (N/%)	82/45.055%
	III (N/%)	1/0.549%
	II/III (N/%)	63/34.615%
	I/II/III (N/%)	4/2.198%
Mechanism of trauma	Scald (N/%)	158/86.813%
	Flame (N/%)	11/6.044%
	Hot oil (N/%)	6/3.297%
	Electrical (N/%)	0/0
	Chemical (N/%)	0/0
	Contact (N/%)	1/0.549%
	Ember (N/%)	6/3.297%
	Inhalation burn (N/%)	2/1.099%
Intervention	Surgery (N/%)	182/100%
	Number of interventions (mean ± SD/range)	2.401 ± 2.002/10
	Skin graft (N/%)	40/21.978%
	Enzymatic debridement (N/%)	6/3.297%
	Fascial excision (N/%)	0/0
	Alloplastic epidermal substitute (N/%)	182/100%

## Data Availability

The original contributions presented in this study are included in the article. Further inquiries can be directed to the corresponding authors.

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
