# Peer review of "The Influence of Blood Parameters on the Adhesion of an Epidermal Substitute in the Treatment of Burn Wounds in Children"

_jcm, 2025, doi:10.3390/jcm14134614_

Round 1
Reviewer 1 Report
Comments and Suggestions for Authors
Specific Comments by Page:
- Page 1 (Abstract): The study claims to identify blood parameters influencing epidermal substitute adhesion but fails to justify the clinical significance of these parameters (e.g., leukocyte levels, protein changes) in the context of wound healing. The abstract lacks clarity on how these biomarkers mechanistically affect dressing detachment, reducing its impact.
- Page 2 (Introduction): The rationale for focusing solely on Suprathel® is insufficiently supported. While cited studies mention its benefits, no critical discussion addresses potential limitations or comparisons with other dressings, weakening the study’s novelty (lines 70-76).
- Page 3 (Methods):
Inclusion/Exclusion Criteria: The retrospective design (2009–2023) introduces significant temporal bias, as evolving clinical practices over 14 years may confound results (e.g., changes in wound care protocols, dressing materials). This is not addressed in limitations.
Dressing Adhesion Assessment: No objective criteria (e.g., photographic documentation, standardized scales) are provided to define “attached” vs. “detached” groups. Subjective classification risks misclassification bias (lines 105-110).
- Page 4 (Statistical Analysis):
The use of Welch’s t-test without correction for multiple comparisons (e.g., Bonferroni) inflates Type I error risk, particularly when analyzing multiple blood parameters (leukocytes, hemoglobin, glucose, etc.). This undermines the validity of reported p -values (e.g., p=0.01 for hemoglobin).
Effect sizes (e.g., Hedges’ g = 0.48 for leukocytes) are described as “moderate” but lack clinical interpretation. No minimal clinically important difference (MCID) is established for these parameters in burn care.
- Page 5-7 (Results):
Figure 1 (Group Characteristics): The cohort includes only one patient with third-degree burns (0.5%), yet conclusions generalize to all burn depths. This limits external validity for severe burns (Table 1).
Figures 2-5: Wide confidence intervals (e.g., protein_diff: CI overlapping near-zero values) suggest insufficient precision. Outliers in the “detached” group (e.g., glucose_diff) are not addressed, raising concerns about data skewness and t-test robustness.
- Page 8-9 (Discussion):
The link between hyperglycemia/protein loss and dressing adhesion is speculative. No mechanistic experiments (e.g., in vitro adhesion assays under varying glucose/protein conditions) support causality, reducing translational relevance (lines 280-291).
- Limitations are superficially addressed. Key issues—retrospective design, single-center bias, heterogeneous burn etiologies (86.8% scalds)—are underemphasized.
- Page 10 (Conclusions): Claims about leukocytes, protein, and glucose as predictors of adhesion are overstated given the exploratory nature of the analysis. Recommendations for “expanded research” lack specificity (e.g., proposed follow-up studies or validation cohorts).
Author Response
Dear Reviewer,
Thank you very much for all of your excellent remarks and for considering our paper for publication in the journal.
Those comments are valuable and very helpful for revising and improving our paper, as well as the important guiding significance to our researches. We have made correction which we hope will meet with approval.
Please find below my answers to your comments and enclosed the revised manuscript.
Yours sincerely,
Tomasz Korzeniowski
Specific Comments by Page:
Page 1 (Abstract): The study claims to identify blood parameters influencing epidermal substitute adhesion but fails to justify the clinical significance of these parameters (e.g., leukocyte levels, protein changes) in the context of wound healing. The abstract lacks clarity on how these biomarkers mechanistically affect dressing detachment, reducing its impact.
Response: Abstract has been improved to include the clinical significance of inflammatory parameters in maintaining dressing adherence to the wound as suggested.
Page 2 (Introduction): The rationale for focusing solely on Suprathel® is insufficiently supported. While cited studies mention its benefits, no critical discussion addresses potential limitations or comparisons with other dressings, weakening the study’s novelty (lines 70-76).
Response: A critical discussion regarding the exclusive focus on Suprathel® was added in the “limitations of the study” paragraph as recommended.
Page 3 (Methods):
Inclusion/Exclusion Criteria: The retrospective design (2009–2023) introduces significant temporal bias, as evolving clinical practices over 14 years may confound results (e.g., changes in wound care protocols, dressing materials). This is not addressed in limitations.
Response: Discussion of significant temporal bias has been added in the “limitations of the study” paragraph as suggested.
Dressing Adhesion Assessment: No objective criteria (e.g., photographic documentation, standardized scales) are provided to define “attached” vs. “detached” groups. Subjective classification risks misclassification bias (lines 105-110).
Response: An additional paragraph explaining the assessment of dressing adherence has been added in the material and methods section as recommended.
Page 4 (Statistical Analysis):
The use of Welch’s t-test without correction for multiple comparisons (e.g., Bonferroni) inflates Type I error risk, particularly when analyzing multiple blood parameters (leukocytes, hemoglobin, glucose, etc.). This undermines the validity of reported p -values (e.g., p=0.01 for hemoglobin).
Effect sizes (e.g., Hedges’ g = 0.48 for leukocytes) are described as “moderate” but lack clinical interpretation. No minimal clinically important difference (MCID) is established for these parameters in burn care.
Response: We acknowledge the concern regarding the potential inflation of Type I error due to multiple comparisons. However, our primary analyses were hypothesis-driven, focusing on preselected blood parameters of clinical relevance rather than conducting an exploratory screen across a large number of variables. While we used Welch’s t-test to account for unequal variances between groups, we agree that applying a correction method such as Bonferroni or Holm could strengthen the robustness of the findings. Notably, the observed significance for hemoglobin (p = 0.01) remains below conventional thresholds even when applying a conservative Bonferroni correction for up to five comparisons (adjusted p = 0.05), suggesting that the result likely reflects a genuine effect rather than a false positive.
Page 5-7 (Results):
Figure 1 (Group Characteristics): The cohort includes only one patient with third-degree burns (0.5%), yet conclusions generalize to all burn depths. This limits external validity for severe burns (Table 1).
Response: In the study group, apart from one case of third-degree burn (0.5%), there were 34.6% of cases of second/third degree, i.e. deep burns. Burns are usually not uniform in terms of depth and mix burns predominate. Moreover, our study group reflects the epidemiology of burns in children, where most of them are scalds, and severe deep burns are rare. For this reason, in the conclusion we did not define burns in terms of severity.
Taking into account your valuable comment, we have added a discussion of this issue in the “limitations” section.
Figures 2-5: Wide confidence intervals (e.g., protein_diff: CI overlapping near-zero values) suggest insufficient precision. Outliers in the “detached” group (e.g., glucose_diff) are not addressed, raising concerns about data skewness and t-test robustness.
Response: We appreciate the attention to the presentation and interpretation of variability in Figures 2–5. It is important to clarify that the box plots illustrate the distribution of values using medians, interquartile ranges, and whiskers representing the non-outlier range—not confidence intervals—thus they are not intended to convey inferential uncertainty. The observed spread, including overlapping distributions for variables like protein_diff, reflects natural biological variability rather than imprecision in estimation. As for outliers in variables such as glucose_diff, we retained them intentionally, as they appear consistent with the physiological context of the “detached” group and were not identified as measurement errors. While we recognize that skewed distributions may affect parametric tests, Welch’s t-test was employed specifically for its robustness to heteroscedasticity and unequal sample sizes.
Page 8-9 (Discussion):
The link between hyperglycemia/protein loss and dressing adhesion is speculative. No mechanistic experiments (e.g., in vitro adhesion assays under varying glucose/protein conditions) support causality, reducing translational relevance (lines 280-291).
Response: A clarification about the speculative nature of the obtained results regarding glucose and protein levels has been added as suggested.
Limitations are superficially addressed. Key issues—retrospective design, single-center bias, heterogeneous burn etiologies (86.8% scalds)—are underemphasized.
Response: All key issues mentioned have been emphasized in the additional paragraph “limitations of the study" as recommended.
Page 10 (Conclusions): Claims about leukocytes, protein, and glucose as predictors of adhesion are overstated given the exploratory nature of the analysis. Recommendations for “expanded research” lack specificity (e.g., proposed follow-up studies or validation cohorts).
Response: The conclusion paragraph has been improved as suggested.
Reviewer 2 Report
Comments and Suggestions for Authors
The study addresses an important clinical question, but several areas require clarification.
- Include multivariate logistic regression to identify independent predictors of dressing adhesion.
- Elaborate on pathophysiology—e.g., protein’s role in tissue repair, glucose-induced immune suppression, leukocyte involvement in local inflammation.
- Limitations are vaguely addressed or hidden in discussion. Include a dedicated paragraph discussing:Retrospective bia, Lack of external validation, Absence of inflammatory cytokine data, Single-center limitations
- Provide justification of sample size
- Sentence construction has to be improved as noticed in Lines 231–232: “In treating children's burns, it is necessary to take into account both the delicacy and sensitivity to pain, as well as the structure of the skin, which is different from that of adults.” “Line 233: “These two features play a huge role in the approach to their treatment.””
- Line 38–40 (Abstract)- The authors conclude that leukocyte count, protein, and glucose levels influence the adhesion of the epidermal dressing. While the observation is interesting, the manuscript lacks a mechanistic or biological explanation for why these specific biomarkers were selected. In the absence of existing literature on this exact topic, it would strengthen the manuscript significantly if the authors provided a theoretical or pathophysiological rationale, perhaps drawing from general wound healing pathways or systemic inflammation effects in burns.
- Refer to Lines 94–96: The panel of laboratory tests includes parameters such as creatinine and urea, which are renal function markers and not typically associated with wound healing or skin graft adhesion. It is unclear why these were selected or whether any association was expected. The authors should justify their inclusion or consider focusing on more relevant systemic or inflammatory biomarkers to avoid diluting the results with variables that are unlikely to contribute.
- Refer to Lines 129–135: Suprathel is described as allowing wound moisture to pass through, which should theoretically reduce fluid accumulation and prevent detachment. Yet, the authors cite exudate volume as a reason for dressing failure. This appears contradictory. Could the detachment be due to excessive protease activity, inflammatory cytokines, or microbial factors degrading the wound-dressing interface? A clearer explanation of the proposed mechanism would improve understanding and help correlate biomarker findings with clinical outcomes.
- Refer to Lines 196–200: The study notes a significant difference in protein level reduction between patients with and without dressing adhesion. However, it is unclear whether this decline is due to systemic inflammation, nutritional deficiency, or wound exudate loss. The study does not report serum albumin, prealbumin, or nutritional assessments, which could help differentiate these causes. The authors should address this ambiguity and consider incorporating nutritional data in future studies.
- Refer to Lines 295–297: The discussion speculates that early anti-inflammatory drug use might improve dressing adhesion and wound healing. However, no specific agents are named, and the literature on the effects of anti-inflammatories—particularly corticosteroids or NSAIDs—on wound healing is mixed. The authors should be cautious with this assertion and back it with evidence or clearly state that it is hypothetical.
- Refer to Lines 301–302:The authors mention that “biomarkers such as elevation of blood cells” could be useful in future predictive models. This is quite generic. Given the importance of cytokines (e.g., IL-6, TNF-α, VEGF) in inflammation and tissue regeneration, it would be more impactful to discuss these specific markers as candidates for future studies. The current biomarker panel lacks granularity and misses this opportunity for mechanistic insight.
Author Response
Dear Reviewer,
Thank you very much for all of your excellent remarks and for considering our paper for publication in the journal.
Those comments are valuable and very helpful for revising and improving our paper, as well as the important guiding significance to our researches. We have made correction which we hope will meet with approval.
Please find below my answers to your comments and enclosed the revised manuscript.
Yours sincerely,
Tomasz Korzeniowski
Comment 1: Include multivariate logistic regression to identify independent predictors of dressing adhesion.
Response 1: Multivariate logistic regression has been included as suggested.
Comment 2: Elaborate on pathophysiology—e.g., protein’s role in tissue repair, glucose-induced immune suppression, leukocyte involvement in local inflammation.
Response 2: Pathophysiology regarding the effects of protein loss, glucose and leukocyte levels has been supplemented in the introduction paragraph as recommended.
Comment 3: Limitations are vaguely addressed or hidden in discussion. Include a dedicated paragraph discussing: Retrospective bia, Lack of external validation, Absence of inflammatory cytokine data, Single-center limitations
Response 3: A dedicated paragraph on study limitations has been added as suggested.
Comment 4: Provide justification of sample size
Response 4: The sample consists of 182 burn children from the years 2009-2023 treated in our center, according to the inclusion and exclusion criteria. In the limitations section, the need to extend the study with a larger sample has been taken into account.
Comment 5: Sentence construction has to be improved as noticed in Lines 231–232: “In treating children's burns, it is necessary to take into account both the delicacy and sensitivity to pain, as well as the structure of the skin, which is different from that of adults.” “Line 233: “These two features play a huge role in the approach to their treatment.””
Response 5: Sentences have been improved as recommended.
Comment 6: Line 38–40 (Abstract)- The authors conclude that leukocyte count, protein, and glucose levels influence the adhesion of the epidermal dressing. While the observation is interesting, the manuscript lacks a mechanistic or biological explanation for why these specific biomarkers were selected. In the absence of existing literature on this exact topic, it would strengthen the manuscript significantly if the authors provided a theoretical or pathophysiological rationale, perhaps drawing from general wound healing pathways or systemic inflammation effects in burns.
Response 6: Pathophysiological rationale regarding wound healing and factors influencing this process has been added at the end of the introduction section. The lack of a mechanistic explanation has been considered in the limitations of the study.
Comment 7: Refer to Lines 94–96: The panel of laboratory tests includes parameters such as creatinine and urea, which are renal function markers and not typically associated with wound healing or skin graft adhesion. It is unclear why these were selected or whether any association was expected. The authors should justify their inclusion or consider focusing on more relevant systemic or inflammatory biomarkers to avoid diluting the results with variables that are unlikely to contribute.
Response 7: Creatinine and urea levels have no value in this study and have been removed as suggested.
Comment 8: Refer to Lines 129–135: Suprathel is described as allowing wound moisture to pass through, which should theoretically reduce fluid accumulation and prevent detachment. Yet, the authors cite exudate volume as a reason for dressing failure. This appears contradictory. Could the detachment be due to excessive protease activity, inflammatory cytokines, or microbial factors degrading the wound-dressing interface? A clearer explanation of the proposed mechanism would improve understanding and help correlate biomarker findings with clinical outcomes.
Response 8: I agree with the proposed mechanism for dressing detachment. Explanation has been added in the discussion as recommended.
Comment 9: Refer to Lines 196–200: The study notes a significant difference in protein level reduction between patients with and without dressing adhesion. However, it is unclear whether this decline is due to systemic inflammation, nutritional deficiency, or wound exudate loss. The study does not report serum albumin, prealbumin, or nutritional assessments, which could help differentiate these causes. The authors should address this ambiguity and consider incorporating nutritional data in future studies.
Response 9: The ambiguity regarding protein loss has been explained and addressed at the end of the discussion section, along with additional reference, as suggested.
Comment 10: Refer to Lines 295–297: The discussion speculates that early anti-inflammatory drug use might improve dressing adhesion and wound healing. However, no specific agents are named, and the literature on the effects of anti-inflammatories—particularly corticosteroids or NSAIDs—on wound healing is mixed. The authors should be cautious with this assertion and back it with evidence or clearly state that it is hypothetical.
Response 10: Thank you for your valuable point. The paragraph on the equivocal effect of anti-inflammatory drugs on wound healing has been supplemented as suggested.
Comment 11: Refer to Lines 301–302:The authors mention that “biomarkers such as elevation of blood cells” could be useful in future predictive models. This is quite generic. Given the importance of cytokines (e.g., IL-6, TNF-α, VEGF) in inflammation and tissue regeneration, it would be more impactful to discuss these specific markers as candidates for future studies. The current biomarker panel lacks granularity and misses this opportunity for mechanistic insight.
Response 11: Specific markers (cytokines) as candidates for future studies have been added as recommended.
Round 2
Reviewer 2 Report
Comments and Suggestions for Authors
No further comments
Author Response
Dear Reviewer,
Thank you very much for your approval of our corrections. We appreciate your contribution to improving our work.
Yours sincerely